# Virulence Genes and In Vitro Antibiotic Profile of *Photobacterium damselae* Strains, Isolated from Fish Reared in Greek Aquaculture Facilities

**DOI:** 10.3390/ani12223133

**Published:** 2022-11-14

**Authors:** Athanasios Lattos, Ioannis A. Giantsis, Eleni Tsavea, Markos Kolygas, Fotini Athanassopoulou, Konstantina Bitchava

**Affiliations:** 1Laboratory of Pathology of Aquatic Animals, Ichthyology & Aquaculture, Faculty of Veterinary Medicine, University of Thessaly, 43100 Karditsa, Greece; 2Environmental Control and Research Laboratory, Region of Central Macedonia, 54625 Thessaloniki, Greece; 3Department of Animal Science, Faculty of Agricultural Sciences, University of Western Macedonia, 53100 Florina, Greece; 4Veterinary Research Institute, Hellenic Agricultural Organization Dimitra, 57001 Thessaloniki, Greece; 5Laboratory of Applied Hydrobiology, Department of Animal Science, School of Animal Biosciences, Agricultural University of Athens, 11855 Athens, Greece

**Keywords:** aquaculture, diseases, *Pasteurella*, *Photobacterium* spp., mortality

## Abstract

**Simple Summary:**

The present study examined and confirmed the presence of *Photobacterium* bacteria in aquaculture units in Greece, where the sector plays a crucial role in the national economy. The majority of these bacteria are characterized by high pathogenicity and antibiotic resistance, indicating the need for appropriate management strategies and systematic surveillance..

**Abstract:**

Bacteria belonging to the species *Photobacterium damselae* are pathogens of cultured marine fish, causing diseases of high importance, such as Pasteurellosis. Thus, they are considered a major threat to the aquaculture sector. Despite the great importance of fish mariculture for the Greek economy, the distribution and abundance of these bacteria are not well documented in aquaculture units in Greece. Keeping this in mind, the scope of the present study was to investigate the presence, antibiotic profile, and virulence of *Photobacterium* bacteria originating from a representative sample of mariculture units throughout Greece. Samples were collected from diseased fish belonging to three different cultured fish species, namely *Sparus aurata*, *Dicentrarchus labrax,* and *Pagrus pagrus,* from both the Aegean and the Ionian Sea. Tissue samples were cultured in agar media, and bacteria were molecularly identified using both bacterial universal and species-specific primer pairs for *Photobacterium* spp. Additionally, the identified strains were characterized for the presence of virulence genes as well as antibiotic profiles. According to the results, the aforementioned bacteria are distributed in the Greek aquaculture units and are characterized by high pathogenicity based on the abundance of virulence genes. Furthermore, the majority of the detected strains exhibit some level of antibiotic resistance. In summary, our results indicate the need for systematic surveillance and study of their antibiotic profiles in Greek aquaculture since these bacteria constitute a major threat to the sector.

## 1. Introduction

The aquaculture sector is one of the fastest-growing sectors in the food industry, providing fish and shellfish for human consumption worldwide [1,2,3]. Although it is considered to be the main supplier of seafood protein for human consumption, intensification of productivity as a result of high demand involves production problems and, in many circumstances, infectious disease outbreaks [4,5]. Occasionally, production losses from aquaculture occur, with infectious diseases being the most important constraint [6]. Disease outbreaks are considered to be a significant threat to the aquaculture sector globally, with an estimation of several billion US$ costs per year, placing large investments at distress risk [7]. Additionally, climate change tends to intensify the outbreaks of infectious diseases in both wild and cultured populations with negative impacts on the cost of aquaculture farms [8].

Infection by *Photobacterium damselae* subsp. *piscicida* (*Phdp*), also known as Pasteurellosis, is a bacterial septic disease hosting a broad range of species in marine and freshwater environments worldwide. It is responsible for huge financial losses in the aquaculture sector [9,10,11]. The chronic form of the disease is characterized by typical whitish tubercules in internal organs, which consist of bacterial accumulations [12,13]. On the other hand, in the acute form of the disease, clinical signs are usually inconspicuous, except for slight hemorrhages on the host [14]. Pasteurellosis has been characterized as “Pseudotuberculosis” [12] due to the high similarity of the clinical signs of the disease with the clinical signs of Mycobacteriosis. It has been demonstrated that stress conditions, such as loss of mucosal layer, skin lesions and low water quality, are implicated in the transmission of the aquatic pathogen though gills, skin and intestine [15]. Antibiotics are the first defense line against pasteurellosis outbreaks in aquaculture [9], however it should be noted that knowledge concerning disease transmission is still scarce. Taking into account the emerging risk of developing resistant strains, current efforts are focused on effective vaccine development to prevent outbreaks of this disease [9].

In the case of infectious diseases, chemotherapeutic practices are included in the confrontation of pathogenic bacteria [16]. For disease prevention and treatment, antibiotics have been intensively used in the aquaculture sector [17]. However, many threats emerged in recent years when these chemotherapeutic agents, in particular antibiotics, are used routinely in off-shore, i.e., fish mariculture, as well as land-based flow through such as rainbow trout and shrimp aquaculture systems, as prevention measurement and not as therapeutic agents [17,18]. Antibiotics can act as inhibitors in bacterial proliferation with a plethora of mechanisms, including inhibition in the cell wall or protein synthesis, suspension in nucleic acid synthesis, and antimetabolite activity, acting at the same time as toxic contaminants in the environment [19,20]. Antibiotics are usually administered as a supplement in the feed or by immersion of the target cultured livestock in closed tanks [20]. Regarding antibiotic usage and maximum residue limits in aquaculture, the use of antibiotics is regulated by the EU council Regulations No 37/2010, No 470/2009, and No 6/2019.

After the use of antibiotics supplementary to feed, a part of antibiotics ends up directly in the environment as non-digested food. Another part is secreted by the fish through feces as a non-absorbed portion, and the absorbed portion of the antibiotics is secreted by the urine and other secretions [21,22,23]. These antibiotic residues accumulate in the surrounding environment (sediments, culture cages) and remain active depending on their significant self-life and other abiotic factors [19,24,25]. Afterward, antibiotic residues can induce antibiotic resistance even in minimum concentrations, promoting the demonstration of antibiotic-resistant bacteria in the culture sites [26,27,28]. The antibiotics licensed for use in Greece intended for fish are: oxytetracycline, florfenicol, flumequine, oxolinic acid, and sulfamethoxazole/trimethoprim and were all used for the antibiotic profiles. 

The use of antimicrobials in every human activity has a direct impact on the environment. Antibiotic resistance is a major problem in various human activities, including aquaculture. Additionally, blind antibiotic treatments induce resistance, and thus the treatment of fish with antibiotics should only be performed after the conduction of an antibiogram.

Additionally, resistant gene transmission between marine and terrestrial bacteria can involve great risks both in public health for the treatment of bacterial infections and in aquaculture for the same reason. It should also be noted that not all bacterial strains are harmful, causing the disease, an incident largely depending on the presence of virulence factors. More specifically, several genes, such as the Apoptosis induced protein gene (*Aip56*), the Adhesion lipoprotein (*pdp-0080*), the periplasmic hemin binding protein (*hutB*), the ABC transporter ATPase (*hutD*), and the Protein 55 gene (*p55)* have been proposed to be associated with virulence of *Photobacterium damselae* subspecies to the fish hosts [29]. 

Greek aquaculture is of extremely high value, and importance for the national economy, is mainly export-oriented. It is placed at one of the first positions in sea bream production worldwide [30,31]. However, despite the crucial role of *Phdp* for the sector, the presence of virulence factors in the genome of strains isolated from Greek marine fish farms has not been well documented. It should be pointed out that Greek aquaculture, in line with global trends, often suffers from disease outbreaks, occasionally not examining in detail the etiological agent. 

Keeping this in mind, the aim of this research was to investigate the presence and molecular identification of *Photobacterium damselae* subspecies in Greek marine aquaculture units alongside the detection of virulence factors among the detected strains hosting various fish species, as well as to estimate their biodiversity and genetic diversity and eventually their susceptibility to commercially used antibiotics and generally the Veterinary Important Antimicrobial Agents (VIA) that are used for all animal species. 

## 2. Materials and Methods

### 2.1. Sampling

Samples were collected from Greek aquaculture farms during outbreaks observed in the period 2019–2021 (Table 1). Most of the Greek farms are located near the coastline. In the area of Thesprotia, the samples were collected from farms that are 150–800 m away from the coastline. This was more than the same for all the other farms from which we isolated the strains. The system in all the farms was intensive, yet the samples were taken from small farms that produce approximately around 500 tons per year. A density of 14 Kgr/m^3^ was mainly observed in sea bream (*Sparus aurata*) and sea bass (*Dicentrarchus labrax*), while for red porgy (*Pagrus pagrus*), it was lower, 9 Kgr/m^3^. All samplings were conducted during Spring and Autumn, with seawater temperatures above 20 °C originating from disease outbreaks. Samples consisted of 8–10 fish per cage, and fish were euthanized by phenoxyethanol overdosing. Fish were examined for the presence of parasites and bacteria in order to find the causative agent of the outbreaks. For the presence of bacteria, samples were collected aseptically from the spleen and anterior kidney of diseased fish. The strains analyzed in this study belong to different cages that had disease outbreaks at different time points. 

### 2.2. Culture Media

Tryptic Soy Agar (TSA, Oxoid) supplemented with 2% NaCl, and blood agar plates were used as non-selective media to isolate any potential microorganism implicated with the disease outbreaks. The plates were incubated for 24–48 h at 25 °C and observed for bacterial growth. For isolation, single colonies were picked and stroked onto new plates. Additionally, Tryptic Soy Broth (TSB, Oxoid) with 2% NaCl was used for the cryopreservation of the strains at −80 °C in a final dilution of 15% glycerol.

### 2.3. Molecular Identification and Phylogenetic Analysis of Photobacterium bacteria

Pure cultures of bacteria stored at −80 °C were incubated onto TSB supplemented with 2% NaCl agar for 24–48 h for the extraction of DNA. DNA extraction was carried out using the DNAEasy Blood and Tissue kit (QIAGEN, Duren, Germany) according to the manufacturer’s guidelines. Extracted DNA quality and quantity were measured in a Q5000 micro-volume UV spectrophotometer (Quawell, China) as well as by electrophoresis in 1.5% agarose gel. For identification purposes, 267 part of 16s rRNA was amplified using the primers CAR-1 and CAR-2 [32] and the FastGene Taq 2x Ready Mix (NIPPON Genetics, Europe) in 20 μL total volume PCRs as well as the generic bacteria primers 27f-CM and 1492r [33] with same volumes and amplification kit. Conditions for CAR1-CAR2 were: 3 min at 95 °C, followed by 38 cycles at 94 °C for 30 s, 51 °C for 40, and 72 °C for 45 s, with a final extension step at 72 °C for 5 min, whereas for 27f-CM-1492r were exactly as in Lattos et al. [34]. Successfully amplified products, according to electrophoresis in 1% agarose gel, were purified using the NucleoSpin Gel and PCR Clean-up kit (Macherey-Nagel, Germany) and sequenced applying the Sanger methodology in an ABI-PRISM 3130xl genetic analyzer using both forward and reverse primer. The software MEGA 7.0 [35] was utilized for editing and analyzing the sequences. Phylogenetic analysis was performed applying the Maximum Likelihood methodology in comparison with two representative sequences, one for *Photobacterium damselae* subsp. *piscicida* (accession number: ON564501) and one for subsp. *damselae* (accession number: MK482016) that were retrieved from the Genbank database after BLAST searches of the newly described sequences.

### 2.4. Detection of the Virulence Factors

Furthermore, for the investigation of the disease pathogenesis of the detected *Photobacterium* spp., the presence of virulence genes was examined as described in Nunez-Diaz et al. [29]. The examined genes were the Apoptosis induced protein (*Aip56*), the Adhesion lipoprotein (*pdp-0080*), the periplasmic hemin binding protein (*hutB*), the ABC transporter ATPase (*hutD*), and the Protein 55 (*p55*). Briefly, extracted DNA from each culture was amplified in five PCRs with primer pairs aip56F-aip56R, pdp-0080F-pdp-0080R, hutBF-hutBR, hutDF-hutDR, and p55F-p55R [29], in 10 μL reaction volumes containing 5 μL FastGene Taq 2x Ready Mix, 0.3 pmol of each primer, 25 ng extracted DNA and water up to the final volume. PCR regime was 95 °C for 3 min, followed by 36 cycles of 94 °C for 30 s, 55 °C for 40 s, and 72 °C for 50 s, and in the end, a final extension step of 72 °C for 7 min. PCR products were visualized in UV light after electrophoresis in a 1.5% agarose gel.

### 2.5. Antibiogram

The susceptibility of isolated strains to various antimicrobials was determined through the disk diffusion method, according to the guidelines of the Clinical and Laboratory Standards Institute [36]. The bacterial strains were inoculated into Mueller-Hinton broth (Oxoid) containing 2% NaCl and incubated overnight at 25 °C. Afterward, each suspension was adjusted to the turbidity of a 0.5 McFarland standard and spread onto Muller-Hinton agar (Oxoid) plates containing 2% NaCl. The antibiograms were prepared using 16 antibiotic disks: florfenicol (FFC, 30 μg), erythromycin (E, 15 μg), cephalothin (KF, 30 μg), cefotaxime (CTX, 30 μg), ampicillin (AMP, 10 μg), amoxicillin/clavulonate (AMC, 20 μg and 10 μg, respectively), kanamycin (K, 30 μg), neomycin (N, 30 μg), gentamycin (GM, 10 μg), streptomycin (S, 10 μg), sulfamethoxazole/trimethoprim (SXT, 1.25 μg, and 23.75 μg, respectively), ciprofloxacin (CIP, 5 μg), flumequine (UB, 30 μg), norfloxacin (NOR, 10 μg) and tetracycline (TE, 30 μg). The antibiotic disks were placed on the plates that were then incubated for 24 h at 25 °C. The diameter of the inhibition zone around each disk was measured and recorded. The results were classified as resistant (R), intermediately resistant (I), or susceptible (S) according to the CLSI guidelines [36]. In some cases, we also used both Mueller-Hinton and blood agar plates for the antibiogram to have a more visible and easier-to-process result (Figure 1). 

## 3. Results

### 3.1. Pathogenicity, Bacterial Culture, Molecular Identification, and Presence of Virulence Factors

During sampling, in most of the cases, the classic symptoms of pasteurellosis were present in the fish, such as splenomegaly, hemorrhages, and whitish tubercules in the spleen and kidney (Figure 2 and Figure 3). The mortalities reported in the farms varied between 5–12%. During the parasitological examination, small numbers of *Sparicotyle chrysophrii* were present in the gills of sea bream and *Diplectanum aequans* in the gills of sea bass, yet the intensity was low, and there were no symptoms (for instance, anemia, necrosis in the gills or weight loss) to assume that the disease was caused by the parasites. No other bacteria were isolated from the fish in those outbreaks. 

All but one strain isolated in TSA agar were identified as *Photobacterium damselae* sbsp. *piscicida* (Table 1) using both applied techniques. Maximum likelihood phylogenetic analysis of the 16S rRNA segment demonstrated only two haplotypes identical with already available *P. damselae* sbsp. *piscicida* and *P. damselae* sbsp. *damselae* haplotypes in the GenBank database (Figure 4, Appendix A). These results confirmed the molecular identification of the cultured bacteria as *P. damselae* sbsp. *piscicida* and *P. damselae* sbsp. *damselae* (Table 1).

### 3.2. Virulence Factors Detected

Additionally, for the detection of the disease pathogenesis, the presence of virulence factors associated with the disease progression was investigated. All strains were positive in the virulence genes *pdp-0080*, *hutB*, *hutD*, and *p55*, with the exception of the *Aip56* gene, which was detected in 14 of 15 strains (Table 2).

### 3.3. Antibiogram

The majority of the isolated strains were susceptible to all the antibiotics tested. Five out of 15 strains, equivalent to one-third of the strains, exhibited some degree of antibiotic resistance to the antibiotics tested. These strains were 228, 407, 400, 417, and 354, and they were isolated from Peloponnisos, Thesprotia, Chalkidiki, and the Aegean Sea, both in the samplings of 2020 and 2021. The antibiotics to which the resistances or intermediate resistances were reported were streptomycin, sulfonamides, ampicillin, novobiocin, tetracycline, and nitrofurantoin (Table 3 and Table 4).

## 4. Discussion

The present study constitutes a generic holistic attempt to investigate the presence of *Photobacterium* bacteria in Greek fish farms, a country of particular importance for the Mediterranean aquaculture industry. In general, the fundamental challenge for aquaculture farms in recent years is the maximization of seafood production with the lowest cost in order to reach the demand for seafood from the constantly growing world population [37]. However, the intensification of the production is accompanied by high-density animal population providing optimal media for disease proliferation and spread [5]. It has been well documented that climate change down-regulates physiology and immunology responses in cultured species shaping them in more vulnerable conditions according to the disease transmission [38,39,40]. Moreover, warm water infectious diseases will tend to outbreak with higher frequency resulting in mortality outbreaks in farms [41]. Regarding opportunistic pathogens, taking advantage of the immunosuppression of the cultured species due to climate change effects, they will have more opportunities to create pathological conditions [42,43]. As for the disease transmission to newly habitat areas, plenty of cases have risen as evidence of the impact of climate change in marine environments, such as the spread of *Perkinsus marinus* and *Haplosporidium nelsoni* in *Crassostrea gigas* [44,45,46]. 

Infections caused by *Phdp* constitute one of the most challenging issues for marine aquacultures contributing to several mortality cases [28]. Pasteurelosis is usually related to temperature increases above 20 °C. However, outbreaks have also been reported in farmed species in temperatures between 18–19 °C [47,48]. According to our results, the vast majority of the examined fish were found positive in the majority of the virulence factors examined (Table 2), indicating the high pathogenicity of both subspecies of *Photobacterium damselae* subspecies hosted in Greek aquaculture units. Concerning the development of the infection, in the early stages, intensive infiltration of macrophages and neutrophils is present in infected tissues [49,50]. Later on, with the progress of the infection, the extensive secretion of the Apoptosis-inducing protein of 56 kDa (AIP56) neutralizing the host phagocyte defense by the apoptotic destruction of macrophages and neutrophils, leads to uncontrollable proliferation of the pathogen and finally in necrotic effects [49,50,51,52,53]. Furthermore, toxin results, as the release of cytotoxic molecules, induce lysis of the host phagocytes, contributing to tissue damage and necrotic lesions [49,54]. Additionally, AIP56 toxicity has been documented to be involved in the downregulation of the transcription factor NF-kB p65 leading to reduced pro-inflammatory cytokine expression [55,56]. Despite the occurrence of the important virulence factor AIP56, adhesion and invasion abilities are essential in *Phdp* infections. Regarding the adhesion of *Phdp*, a lipoprotein (PDP_0080) plays an important role in the adherence of the bacterium to epithelial cells [57]. Concerning the expression of virulence factors contributing to the invasion mechanisms of the bacterium to host cells, there are no factors relating directly to the invasion mechanisms of the bacterium [58,59]. Bacteria need iron for their survival, and they have developed pathways in order to obtain it from their hosts. Bacteria can obtain iron through transferrin, siderophores, or heme [60]. *Phdp* can acquire iron from hemin and hemoglobin through an iron uptake system, including a TonB-dependent outer membrane receptor, a periplasmic binding protein, and an ATP binding cassette system (ABC) [61]. Iron uptake of *Phdp* from the host can be performed by synthesis and the transfer of siderophore piscibactin in the cell through the outer membrane receptor FrpA [7,62].

Furthermore, it is known that *Phdp* possesses a gene cluster structure implicated in the biosynthesis of a piscibactin, and a plasmid of this cluster can be transferred to other gram-negative bacteria such as *Vibrio alginolyticus*, facilitating them to grow under iron-limited environment [6,61]. Acquired resistance of bacteria to antibiotics is a biological phenomenon owing to either random mutations in the genome of the microorganism, which are transmitted vertically to subsequent generations, or the acquisition of exogenous genetic elements such as plasmids, which carry genes that confer resistance. Concerning the presence of virulence genes in the bacterial colonies isolated from outbreaks in our study, the samples demonstrated almost all the virulence genes selected for analysis. Specifically, the presence of HutB genes that their participation is related to the coding of heme transfer proteins to the ABC transporter from the receptor demonstrated in all bacterial samples implicated in this research [59]. This indicates a relationship with the rest of the studies, demonstrating an ability to exploit iron from host tissue cells that are mandatory for bacterial pathogenicity. HutD gene presence was also confirmed in all strains, and its implication in iron uptake had been documented both in *Phdp* and *Vibrio* spp. in previous studies [63,64]. Regarding the p55 gene, whose role has not been clarified completely, it was also detected in all *Phdp* strains. Despite the lack of knowledge about its involvement in the pathogenicity of *Phdp*, homologues of this protein has been implicated in several gram-negative bacterial proteins and documented in the *Mycobacterium marinum* virulence during the infection in *Danio rerio* [65,66]. Concerning the adhesion of the pathogen in the host cells, lipoprotein pdp_0080 has been reported to be involved in the function [48]. The presence of this virulence factor was detected in all samples of this research, confirming the mandatory role of this protein in the pathogenicity of the bacterial species. Despite their role in the adhesion, lipoproteins exhibit an important role in the interactions between pathogen and host in adhesion, translocation, and evading ability in the immune system of the host [29,67,68]. AIP56 is an AB toxin secreted by *Phdp* and constitutes a main virulent factor [51]. The presence of this key virulent factor was detected in 14 of 15 samples. Strain 355 didn’t contain this gene, yet we are not able to conclude if the strain (355) is virulent or not. It has been stated that the absence of AIP56 does not necessarily render the strain non-infectious [53].

In order to treat the disease after the pathogen invasion in the cultured animal stock, the use of antibiotics is still playing a crucial role, despite the concerns about the transport of the antibiotic resistance genes between bacterial populations [69]. However, the discovery of antibiotics was an important tool for saving livestock from infectious diseases [70]. The problem of indiscriminate use of antibiotics constitutes a serious global threat, and alternative therapies against multidrug-resistant bacteria are urgently needed to treat bacterial infectious diseases in the future [71]. Antibiotics have been used in aquaculture for more than 50 years, killing bacterial populations or inhibiting their growth [72]. Although the development of vaccines in recent years, alongside the optimization of management practices, has led to a great reduction in antibiotic use, antibacterial therapies are still the last option to deal with bacterial infections [73,74,75,76].

Of the strains analyzed in this study, the majority were sensitive to most of the antibiotics tested. Yet, five out of 15 strains, equivalent to one-third of the strains, exhibited some degree of antibiotic resistance to the antibiotics tested (228, 407, 400, 417, 354). These strains were isolated both in 2020 and 2021 and originated from almost all the sampling locations representative of the wider area of fish mariculture in Greece, as reported in Table 1. Resistant bacterial strains were reported in streptomycin, sulfonamides, ampicillin, and novobiocin. In particular, three strains were resistant to streptomycin, and two strains to sulfonamides, ampicillin, and novobiocin. Resistance to tetracycline and nitrofurantoin was only reported in one strain, respectively. Considering the antibiotics used in the Greek aquaculture industry, it must be pointed out that these are tetracyclines, sulfonamides, and quinolones (flumequine, florfenicol, and oxolinic acid). According to Thyssen and Ollevier [77], who studied the in vitro antimicrobial susceptibility of *Phdp* strains isolated from different countries, the difference in availability of drugs from every country corresponds to an increased possibility of developing resistances to the antimicrobials most frequently used. In our study, there were resistances developed in 2 strains for sulfonamides and in one strain for tetracycline, although most of the resistances were to antimicrobials not used in the aquaculture industry. This could be a result of the transmission of resistant genes from environmental bacteria non-pathogenic to fish species to pathogenic ones as a result of the indiscriminate use of antibiotics by humans. 

## 5. Conclusions

*Photobacterium damsealae* pathogenic strains were isolated both in the Ionian and Aegean Sea in three different Mediterranean cultured fish species. The distribution of these strains in Greek aquaculture indicates their importance as pathogens and the need to apply more effective measures to combat them. Systematic surveillance at early fish stages and routine sampling, together with the use of vaccines in all fish sizes and species, could be a strategy to prevent undesired phenomena that threaten the viability of the sector. The presence of *P. damselae* sbsp. *damselae* should be further studied since there are no commercial vaccines for this species. The scarcity of appropriate health management in the farms results in the incorrect diagnosis and ineffective usage of antibiotics due to the inability to perform antibiograms. This finally leads to the excessive use of antibiotics and to the emergence of antibiotic resistance. 

## Figures and Tables

**Figure 1 animals-12-03133-f001:**
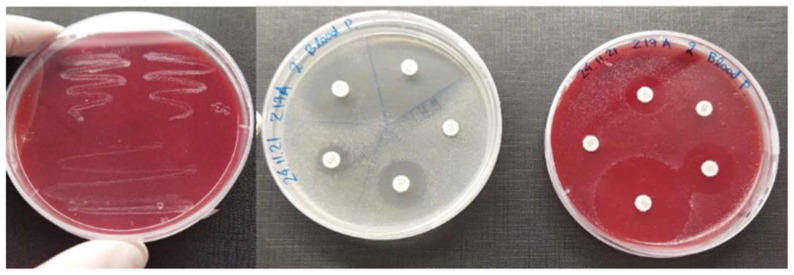
Culture of *P. damselae* sbsp. *piscicida* strain in blood agar plate (**upper**). Picture of an antibiogram in Mueller Hinton (**lower**, **left**) and blood agar plate (**lower**, **right**).

**Figure 2 animals-12-03133-f002:**
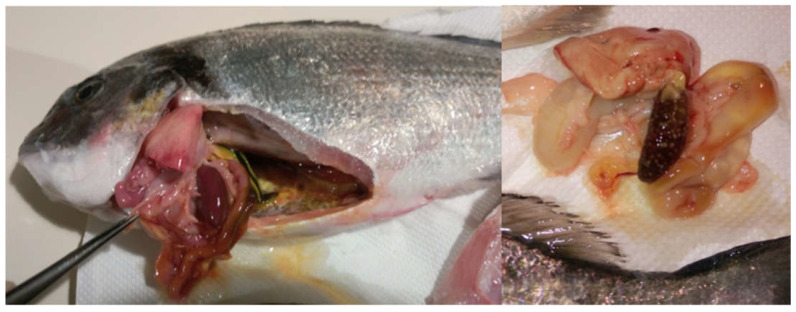
Splenomegaly and congestion in sea bream (*Sparus aurata*) (**left**), and whitish tubercules in the spleen of diseased sea bream (**right**).

**Figure 3 animals-12-03133-f003:**
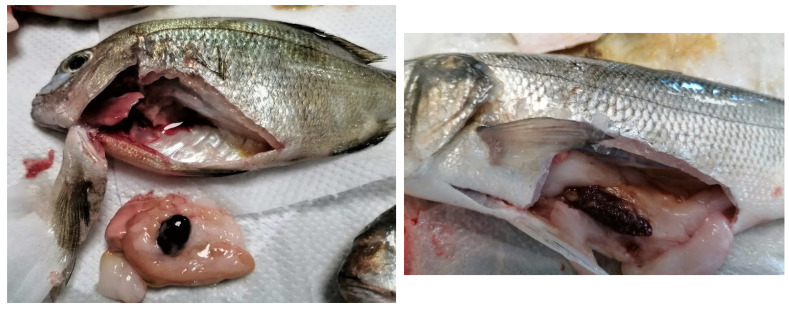
Splenomegaly in red porgy (*Pagrus pagrus*) (**left**) and sea bass (*Dicentrarchus labrax*) (**right**).

**Figure 4 animals-12-03133-f004:**
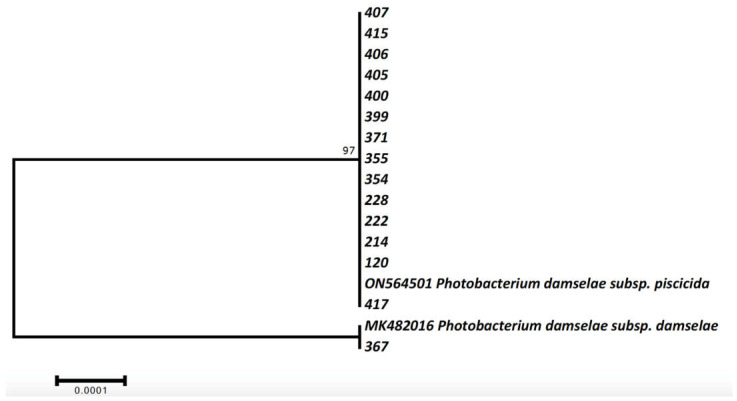
Maximum likelihood phylogenetic tree of the cultured bacteria strains. Codes as in Table 1.

**Table 1 animals-12-03133-t001:** Code, identification, seawater temperature, location, host, and collection period of the isolated strains.

*Sampling Code*	Molecular Identification	Collection Period	Seawater Temperature (°C)	Location	Host
222	*P. damselae* sbsp. *piscicida*	October 2019	24	Thesprotia	sea bass
407	*P. damselae* sbsp. *piscicida*	November 2021	21	Thesprotia	red porgy
354	*P. damselae* sbsp. *piscicida*	October 2020	22	Chalkidiki	sea bass
415	*P. damselae* sbsp. *piscicida*	November 2021	21	Thesprotia	red porgy
120	*P. damselae* sbsp. *piscicida*	October 2019	25	Evia	sea bass
214	*P. damselae* sbsp. *piscicida*	October 2019	22	Chalkidiki	sea bream
355	*P. damselae* sbsp. *piscicida*	October 2020	25	Thesprotia	red porgy
367	*P. damselae* sbsp. *damselae*	October 2020	24	Thesprotia	sea bass
228	*P. damselae* sbsp. *piscicida*	December 2020	21	Aegean Sea	sea bream
399	*P. damselae* sbsp. *piscicida*	November 2021	23	Peloponnisos	sea bass
406	*P. damselae* sbsp. *piscicida*	November 2021	23	Peloponnisos	sea bass
400	*P. damselae* sbsp. *piscicida*	November 2021	22	Peloponnisos	sea bass
371	*P. damselae* sbsp. *piscicida*	December 2021	17	Thesprotia	sea bream
417	*P. damselae* sbsp. *piscicida*	November 2021	23	Peloponnisos	sea bream
405	*P. damselae* sbsp. *piscicida*	November 2021	20	Thesprotia	red porgy

**Table 2 animals-12-03133-t002:** Detection of virulence genes in the cultured bacteria. Codes as in Table 1.

*Photobacterium damselae* sbsp. *Piscicida*	Virulence Genes				
Number of Isolated Strain	*Aip56*	*pdp-0080*	*hutB*	*p55*	*hutD*
222	+	+	+	+	+
407	+	+	+	+	+
354	+	+	+	+	+
415	+	+	+	+	+
120	+	+	+	+	+
214	+	+	+	+	+
355	-	+	+	+	+
367	+	+	+	+	+
228	+	+	+	+	+
399	+	+	+	+	+
406	+	+	+	+	+
400	+	+	+	+	+
371	+	+	+	+	+
417	+	+	+	+	+
405	+	+	+	+	+

‘’+’’ indicates the presence of the target gene, while ‘’-‘’ indicates the absence of the target gene.

**Table 3 animals-12-03133-t003:** List of the strains and the antibiotics used for the antibiograms (CTX: cefotaxime, AMP: ampicillin, AX: amoxicillin, S: streptomycin, GM: gentamicin, S: streptomycin, E: erythromycin, SXT: sulfamethoxazone/trimethoprim, FFC: florfenicol. The results are depicted by R: resistant, S: susceptible, and I: intermediate.

	Cephalosporin	Penicillins		Aminoglycosides		Macrolide	Sulfonamide	Phenicol
STRAIN	CTX	AMP	AX	S	GM	E	SXT	FFC
120	**S**	**S**	**S**	**S**	**S**	**S**	**S**	**S**
214	**S**	**S**	**S**	**S**	**S**	**S**	**S**	**S**
222	**S**	**S**	**S**	**S**	**S**	**S**	**S**	**S**
228	**S**	**S**	**S**	**R**	**S**	**S**	**R**	**S**
354	**S**	**R**	**S**	**S**	**S**	**S**	**S**	**S**
355	**S**	**S**	**S**	**S**	**S**	**S**	**S**	**S**
367	**S**	**S**	**S**	**S**	**S**	**S**	**S**	**S**
371	**S**	**S**	**S**	**S**	**S**	**S**	**S**	**S**
399	**S**	**S**	**S**	**S**	**S**	**S**	**S**	**S**
400	**R**	**S**	**S**	**I**	**S**	**S**	**S**	**S**
405	**S**	**S**	**S**	**S**	**S**	**S**	**S**	**S**
406	**S**	**S**	**S**	**S**	**S**	**S**	**S**	**S**
407	**S**	**R**	**S**	**I**	**S**	**S**	**S**	**S**
415	**S**	**S**	**S**	**S**	**S**	**S**	**S**	**S**
417	**S**	**S**	**S**	**S**	**S**	**S**	**I**	**S**

**Table 4 animals-12-03133-t004:** List of the strains and the antibiotics used for the antibiograms (CIP: ciprofloxacin, ENR: enrofloxacin, TE: tetracycline, NV: novobiocin, NA: nalidixic acid, RD: rifampicin, 0/129: vibriostatic). The results are depicted by R: resistant, S: susceptible, and I: intermediate.

		Fluoroquinolones		Tetracycline	Aminocoumarin	Quinolones	Antimycobacterials		
STRAIN	SPECIES	CIP	ENR	TE	NV	NA	RD	O/129	Nitrofurantoin (F)
120	*P. damselae* subsp. *piscicida*	**S**	**S**	**S**	**S**	**S**	**S**	**S**	**S**
214	*P. damselae* subsp. *piscicida*	**S**	**S**	**S**	**S**	**S**	**S**	**S**	**S**
222	*P. damselae* subsp. *piscicida*	**S**	**S**	**S**	**S**	**S**	**S**	**S**	**S**
228	*P. damselae* subsp. *piscicida*	**S**	**S**	**I**	**I**	**S**	**S**	**S**	**S**
354	*P. damselae* subsp. *piscicida*	**S**	**S**	**S**	**S**	**S**	**S**	**S**	**S**
355	*P. damselae* subsp. *piscicida*	**S**	**S**	**S**	**S**	**S**	**S**	**S**	**S**
367	*P. damselae* subsp. *damselae*	**S**	**S**	**S**	**S**	**S**	**S**	**S**	**S**
371	*P. damselae* subsp. *piscicida*	**S**	**S**	**S**	**S**	**S**	**S**	**S**	**S**
399	*P. damselae* subsp. *piscicida*	**S**	**S**	**S**	**S**	**S**	**S**	**S**	**S**
400	*P. damselae* subsp. *piscicida*	**S**	**S**	**S**	**S**	**S**	**S**	**S**	**S**
405	*P. damselae* subsp. *piscicida*	**S**	**S**	**S**	**S**	**S**	**S**	**S**	**S**
406	*P. damselae* subsp. *piscicida*	**S**	**S**	**S**	**S**	**S**	**S**	**S**	**S**
407	*P. damselae* subsp. *piscicida*	**S**	**S**	**S**	**I**	**S**	**S**	**S**	**R**
415	*P. damselae* subsp. *piscicida*	**S**	**S**	**S**	**S**	**S**	**S**	**S**	**S**
417	*P. damselae* subsp. *piscicida*	**S**	**S**	**S**	**S**	**S**	**S**	**S**	**S**

## Data Availability

Data is contained within the article and Appendix A.

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
