# Peer review of "Virulence Genes and In Vitro Antibiotic Profile of *Photobacterium damselae* Strains, Isolated from Fish Reared in Greek Aquaculture Facilities"

_animals, 2022, doi:10.3390/ani12223133_

Round 1

Reviewer 1 Report (Previous Reviewer 2)

I would like to thank authors for preparing the new version of the manuscript. 

Please find some comments:

1) line 159: Piscicida should be italic

2) Table 2: name of virulence genes should be in italic.

Please correct all of them during the manuscript.

Reviewer 2 Report (Previous Reviewer 3)

This is a good study. It would be better if more samples could be added.

This manuscript is a resubmission of an earlier submission. The following is a list of the peer review reports and author responses from that submission.

Round 1

Reviewer 1 Report

MANUSCRIPT NUMBER: Manuscript ID: animals-1915180

MANUSCRIPT TITLE: Virulence genes and in vitro antibiotic profile of Photobacterium damsealae strains, isolated from fish reared in Greek aquaculture facilities (the name of the specie is not correct in the title)

GENERAL COMMENTS:

This is a very important issue for aquaculture, especially in intensive production. In the manuscript was important to have information about the type of production systems, fish intensity, between others. General characterization of the production areas it is also needed because the antibiotics resistance cannot exclude an anthropogenic origin depending on the nearness of the coast. It is important to well characterize the production system and dissociate the regular use of antibiotics mentioned in this work from all the production systems, especially when we talk about routine and preventive treatments. Such as the semi-intensive systems typical in Mediterranean area where the use regular of antibiotics is not a common practice.

Another issue that should be part of the introduction is antibiotics that are used in aquaculture. 15 antibiotics are tested and this should be clarified even from the point of view that it is authorised by the EU.

SPECIFIC COMMENTS:

Line 79 – Delete Thee point after pagrus

Line 31 – spp. because the Genera represents more than one specie (check all the document)

Line 38 – The same for sp.

Line 52 – And the abbreviation name Phdp after the specie name

Line 65 – Clarify for aquaculture the “used routinely as preventive measurement” what system, offshore? Species? Intensive production. Are the farms near the coast?

Line 107 – anterior kidney?

Line 119 – A spectrophotometer do not allow an efficient measurement of samples quality. Although the ratio could give an indication of the purity it is recommended to visualize it by electrophoresis.

Line 177 – In the table 1, the code number isn´t an important information for the manuscript. But is missing the fish weight and for example the associated mortality, number of fish’s sampled. Also, the details of biological sampling, how you killed the fish for sampling. The data from the production system (cages? Offshore), the density of production etc are necessary.

Also in table 1, It may be interesting to organize the information by location.

Figure 1 - and all figures – add the scientific name of the species.

Line 180- AddAddicionally…correct

Line 183 – Strain 14 and 15????

In table 2- Y and – means? in the table. Also,I would advise to include the result of the PCR (agarose gel) as supplementary data.

Line 193- Is missing the subtitle of the Table 3.And the codes  CtX AMP (meaning in the subtitle)

Line 195- It is Table 4. And rewrite the subtitle and the information about the organization of the antibiotics.

Line 218- Reference 12, add more recent bibliography.

Line 306-310- This issue should be addressed in the introduction. This is not a conclusion from this manuscript.

Line 311-316- This manuscript does not show that there is an increase of bacteria antibiotics resistance to of these bacteria, nor it is a temporal work on this topic. Rewrite the conclusions. Also, it is very dangerous to bring these kinds of conclusions, which are in the future taken into account in EU legislation and other rules. The author must be very careful and conclude only what the work is about.

Reviewer 2 Report

I was honored to review the manuscript entitled “Virulence genes and in vitro antibiotic profile of Photobacterium damsealae strains, isolated from fish reared in Greek aquaculture facilities” submitted to Animals Journal. Photobacterium damsealae subspecies piscicid and damsealae are causative agents of Pasteurellosis and vibriosis respectively. This research article could help to open the wide insight about the virulence genes and the effect of some antibiotics against Pasteurellosis. I would like to thank authors for preparing this research article in the perfect way. This study has an acceptable quality and well written. However, it needs some correction for improving the quality. Here are some grammatical and scientific  points that I suggested:

1) Line 104: Which methods did you use for detecting the other diseases?

2) Please add information regarding to treatment and prevention of this disease in introduction. Moreover, information about transmission.

3) The number of samples is not clear. How many fish samples did you take?

4) Table 1: As far as I know, P. damsealae sbsp. Damsealae is causative agent of vibriosis however, there is no information about this disease. Please clarify.

5) Which methods of sequencing did you use? Illumina or Sanger? As well as add sequencing data in supplementary data.

6) Please add information about detection of virulence factors in details.

7) Please add image of media and antibiogram test in as a supplementary data.

8) Line 180: Additionally

9) Table 3: what is CN antibiotic? Do you mean GM (Gentamicin)?

10) In material and methods has been mentioned 15 antibiotics however, in table 3 is 16 antibiotics. Please clarify

11) Please merge table 3 in one page.

12) Line 182: Aip56 should be italic also in table 2. Please correct during the text.

13) Please rewrite line 291-292

In conclusion I believe that the present manuscript can be accepted after major revision, for publication.

Reviewer 3 Report

This study was to investigate the distribution, abundance, antibiotic profile and virulence of Photobacterium bacteria originating from a representative sample of mariculture units throughout Greece. As an paper studying epidemiology, it need to collect more representative samples of Photobacterium bacteria. If there are no more strains of Photobacterium, it is suggested that the authors can supplement the regression infection experiment of Photobacterium damsealae to verify that those the classical symptoms are actually caused by Photobacterium damsealae.

Above are my main concerns, there are some minor issues also need to be addressed:

1. The Photobacterium damsealae came from the sea bream (Sparus aurata), sea bass (Dicentrarchus labrax) and red porgy (Pagrus pagrus). It is recommended to supplement clinical photos of disease fish.

2. Table 3 needs to be rearranged.

3. Line 142:the FastGene Taq 2x Ready Mix as described in 2.2→2.3

4. From the sensitivity results in this article it appears that 1/3 of the bacteria are resistant to a few antibiotics. So, it is not possible to conclude that the majority of the detected strains exhibit high levels of antibiotic resistance.

5. Line 142: 10 μl reaction volumesOr 100 μl.